# Job Insecurity in Nursing: A Bibliometric Analysis

**DOI:** 10.3390/ijerph18020663

**Published:** 2021-01-14

**Authors:** Vicente Prado-Gascó, María del Carmen Giménez-Espert, Hans De Witte

**Affiliations:** 1Department of Social Psychology, Faculty of Psychology, University of Valencia, 46010 Valencia, Spain; vicente.prado@uv.es; 2Department of Nursing, Faculty of Nursing and Chiropody, University of Valencia, 46010 Valencia, Spain; 3Research Group Work, Organizational and Personnel Psychology, Faculty of Psychology and Educational Sciences, KU Leuven, B-3000 Leuven, Belgium; hans.dewitte@kuleuven.be; 4Optentia Research Focus Area, North-West University, 1900 Vanderbijlpark, South Africa

**Keywords:** bibliometric analysis, job insecurity, nursing

## Abstract

Nurses are a key workforce in the international health system, and as such maintaining optimal working conditions is critical for preserving their well-being and good performance. One of the psychosocial risks that can have a major impact on them is job insecurity. This study aimed to carry out a bibliometric analysis, mapping job insecurity in 128 articles in nursing, and to determine the most important findings in the literature. The search was conducted in the Web of Science Core Collection database using the Science Citation Index (SCI)-Expanded and Social Sciences Citation Index (SSCI) indexes on 6 March 2020. This field of discipline has recently been established and has experienced significant growth since 2013. The most productive and widely cited authors are Denton and Zeytinoglu. The most productive universities are Toronto University, McMaster University, and Monash University. The most productive countries are the United States, Canada, Australia, Finland, and the United Kingdom. The most widely used measure was Karasek’s Job Content Questionnaire (JCQ). The main findings report negative correlations with job satisfaction, mental well-being, and physical health. Job insecurity is a recent and little-discussed topic, and this paper provides an overview of the field. This will enable policies to reduce psychosocial risks among nurses to be implemented.

## 1. Introduction

The restructuring of labor markets, economic crisis and globalization, and increasingly excessive labor demands on workers, (flexible contracts, reduced staffing, temporary contracts, increased workload and pressure, and poor work-life balance), can become labor stressors throughout the world in both industrialized and developing countries [1]. However, work-related stress is determined by multiple causes [2], including psychosocial risk factors, namely work design characteristics related to the general context of the organization that may impair the psychological and/or physical well-being of employees [3]. Among the psychosocial risks that seem to negatively affect workers, job insecurity is one which has the most impact on workers and on society in general [4]. Job insecurity can be defined as a concern about continuing future employment [5] and is a subjective perception: workers may perceive the same situation in slightly different ways due to their personality or position in the labor market [6]. Job insecurity is a work stressor, with negative consequences for the individual’s physical, psychological and social health [6,7], and their family [8]. Theories such as the Karasek’s Job Demand Control Model [9] contribute to the interpretation of job insecurity. This model explains job stress in terms of the balance between the psychological job demands and the worker’s level of control over them, i.e., job control [9]. Psychological job demands include role conflict, workload, role ambiguity, job insecurity, and cognitive demands, among others, and are the psychological stressors in the work environment. Job control involves the opportunities to develop one’s own skills, and the autonomy that the work provides, i.e., the resources that the employee has. The development of these skills has a two-fold aspect; on the one hand obtaining and improving sufficient capacity to carry out the tasks, and on the other hand, the possibility of working on (or carrying out) what one knows best (creative and varied work). Autonomy refers to the ability to decide on one’s own tasks and control over breaks and the pace of work [9]. According to Kasarek, the Job Demand Control Model predicts that low levels of job control, high demands at work and the interaction between them lead to a higher risk of strain, including poorer health and well-being [10]. Job insecurity is described in terms of lack of control, and implies a lack of resources, especially control [10]. In this model, the health or well-being of the worker will depend on the balance between the demands of the job and the worker’s own available resources. The authors Johnson and Hall introduced social support as the third dimension of this model, by establishing the job demand-control-social support model, operating in two ways [11] since workers exposed to high demands, little control and under social support present twice the risk of morbidity and mortality from cardiovascular disease than those with low-demand jobs, much control and strong social support. This dimension of social support refers to all possible levels of on-the-job interaction, both with peers and superiors.

Job insecurity is the most important aspect of work in almost all countries according to the Organisation for Economic Co-operation and Development (OECD), and the European Social Survey (ESS) and the International Social Survey Programme (ISSP) confirm this perception among workers [12]. In addition, job insecurity not only has adverse effects on people’s health and well-being; it also negatively affects employees’ job satisfaction and commitment, and reduces their work-related health and well-being [4,13]. Job insecurity also reduces the satisfaction of basic human needs [9], performance at work, and creativity [14,15]. Furthermore, it can affect companies’ performance by reducing worker retention rates, investment in company-specific skills and productivity, and impacts on society at large, social unrest, consumer confidence, and savings behavior. For these reasons, insecurity is one of the three main dimensions in the new OECD framework for the measurement and assessment of the quality of employment [16].

These aspects are crucial to any type of profession but are paramount for nurses, because nurses are the largest group of health professionals (59%) and play a vital role in global health systems [17]. The contribution of nurses to global health is undisputed, and investing in improving their quality-of-life benefits society [18]. Job insecurity among nurses has also been associated with migration to other countries [19] in search of higher salaries and a better quality of life. It has also been linked to worsening health conditions for nurses and increasing burnout, stress and vulnerability [20,21], lower job satisfaction levels [22]. The increase in nurses’ workload increases the probability of hospital deaths, and significantly reduces the quality of care [23] and holistic nursing care [24]. Job insecurity makes daily work and therapeutic relationships difficult [25] because professionals are unable to provide quality care. Improved working conditions and professional development affects not only their well-being and quality of life, but also their performance and by extension the functioning of the entire healthcare system [26]. In addition, an increase in demand for nursing care and the shortage of nurses worldwide is a concern, and adequate prioritization of occupational health and safety is essential as a result, as the World Health Organization (WHO) suggests [17].

Despite its importance, the decline in the average employment tenure of people aged 30–50 in all countries between 1992 and 2014 is evidence of increasing job insecurity [27]. The existing literature on this subject is scarce, and a bibliometric study on the subject in nursing appears to be lacking. For this reason, this study analyzes the state of research on job insecurity in nursing using a bibliometric analysis to determine the most important findings in the literature in this field, taking into account the distribution of publications, authorship, co-authorship, impact, the most prolific institutions and countries, citation networks, and the most relevant topics about job insecurity in nursing [28]. It also aims to establish the most widely used instruments for measuring job insecurity, and the main causes and consequences of job insecurity among nurses [29]. As a result of the lack of studies of this type, we believe that it could be very useful for health policymakers and health managers in all countries and regions to design programs to retain nurses and ensure the sustainability of health care systems [19], improve nursing processes and working conditions [30].

## 2. Materials and Methods

A descriptive bibliometric analysis of job insecurity in nursing was carried out. Bibliometric studies explore publishing patterns and trends, describe concept development, new emerging areas of research, research gaps, and information about and characteristics of the existing literature and recent advances [31]. The analysis and bibliometric maps were conducted using Thompson Reuters’ Journal Citation Reports (JCR^®^) [32], identifying authors, institutions, and countries that have significantly contributed to the development of a specific area of research. The search was conducted in the Web of Science Core Collection database using the SCI-EXPANDED and SSCI indexes on 6 March 2020. The broadest possible topics related to job insecurity in the nursing context were used. The title, abstract, and keywords were used to identify contributions related to this topic from various perspectives. The search strategy included the most significant theoretical concepts: job insecurity (8,272) and nursing (208,161). The search syntaxis used was as follows: TS = (“*EMPLOY* *CERTAIN*” OR “*EMPLOY* *SECURIT*” OR “CAREER *CERTAIN*” OR “CAREER *SECURIT*” OR “CAREERS *CERTAIN*” OR “JOB* *CERTAIN*” OR “JOB* *SECURIT*” OR “LABOR *CERTAIN*” OR “LABOR *SECURIT*” OR “LABOUR *CERTAIN*” OR “LABOUR *SECURIT*” OR “MÉTIER *CERTAIN*” OR “MÉTIER *SECURIT*” OR “OCCUPATION* *CERTAIN*” OR “OCCUPATION* *SECURIT*” OR “PROFESSION* *CERTAIN*” OR “PROFESSION* *SECURIT*” OR “WORK *CERTAIN*” OR “WORK *SECURIT*”) OR TS = (*EMPLOY* OR JOB* OR LABOR* OR LABOUR* OR MÉTIER*)) AND (TS = (*CERTAIN* OR *SECURIT*)) AND (TS = (NURS* OR NURSING)).

The search generated 254 articles that were reviewed for duplicate records, as well as their suitability according to the inclusion criteria considered. The inclusion criteria used to determine the studies in this research were: (1) literature reviews and empirical studies, (2) scientific journal articles, (3) published in any language in all years, (4) in the Web of Science Core Collection SCI EXPANDED and SSCI database (5) examining job insecurity in nurses in any care setting (hospitalization, special services, home care, primary care). In the first step, reading the titles and abstracts of the articles led to the exclusion of 100 articles that did not meet the inclusion criteria. In the second step, a complete reading of the remaining 154 articles led to the exclusion of 26 articles that did not meet the inclusion criteria, leaving a final sample consisting of 128 articles from the WoS from between 1993 and 2020. The quantitative content analysis was implemented using a bibliometric analysis and mapping 128 articles on job insecurity in nursing. We then reviewed the homonymy and synonymy in the authors’ names. We used the author’s affiliation available in the WoS database, and an additional Google search was performed in its absence [33]. The removal of these ambiguities is necessary since they impair bibliometric analyses such as co-authorial link predictions [34], collaborative network analysis [35], and citation network analysis [36].

The data was analyzed in three phases. A bibliometric analysis and mapping was performed in the first phase. The programs used were Hiscite (version 2012.03.17; HistCite Software LLC, New York, USA), Bibexcel (version 2016.02.20; Olle Persson, Umeå University, Umeå, SWE), Pajeck (version 5.06, 2013.11.12; Batagelj and Mrvar, University of Ljubljana, Ljubljana, Slovenia) and Vosviewer (version 1.6.9, Eck and Waltman (2013), Leiden University, the Netherlands). The HistCite program (version 2010.12.6) permits identification of significant articles in WoS topic searches, contributing to the bibliometric analysis. The Global Citation Scores (GCS) were acquired using HistCite. The knowledge maps showing the frequency of occurrence and the relationships between terms were constructed using Bibexcel (version 2011.02.03) in combination with Pajeck (version 3.14) and Vosviewer, providing the display of the different bibliometric maps. The following data were extracted from each article in the second phase: study citation, type of study (empirical study, systematic review, meta-analysis), objective, job insecurity (data prevalence, intervention), main causes and consequences of job insecurity, measuring instruments used, and financing. Finally, in the last phase, the articles in which job insecurity is a central aspect of the study and not just one of various aspects to be evaluated within a psychosocial risk analysis perspective (13 papers) were analyzed in depth (Figure 1). The information from phases two and three were collected in a Microsoft Excel^®^ database created ad hoc.

## 3. Results

### 3.1. Main Bibliometric Indicators

The first article was published in 1993, and there was significant growth by 2013, when 11 articles were published, followed by 2014 and 2015 when 11 articles were published each year. Finally, 10 articles were published in 2016. There was a decline in 2017 and 2018, with 5 and 7 articles, respectively. Twelve articles were published in 2019, and 2 articles have been published in 2020 (until March 6). The number of articles published per year ranged from 1 to 12 (Mean = 4.92, SD = 3.53); the overall citation scores (GCS) per year ranges from 0 to 181 (Mean = 76.58, SD = 51.51). The year with the highest number of citations was 2013 (GCS = 181), followed by 2005 (GCS = 154) and 2001 (GCS = 146).

The number of articles published per journal ranged from 1 to 10 (Mean = 1.42, SD = 2.21) the Journal of Advanced Nursing (N = 10 articles) and the Journal of Nursing Management (N = 6 articles) were the most productive journals, followed by the International Journal of Nursing Studies and the Journal of Nursing Administration (4 articles each). The remaining journals published 3 or fewer articles. The citations ranged from 0 to 278 (Mean = 22.12, SD = 35.68). The results indicate higher values for the Journal of Advanced Nursing (GCS = 278), followed by Work and Stress (GCS = 111) and the International Journal of Health Service (GCS = 96). The remaining journals received 78 citations or less. The GCS per year (GCS/t) ranged from 0 to 21.44 (Mean = 2; SD = 1.80).

The articles originated in 157 different countries (range 1–21, Mean = 4.03, SD = 4.62). The most productive countries were the United States (N = 21 articles) and Canada (N = 19 articles), followed by Australia, Finland, and the United Kingdom (N = 10 articles each). Next were China (N = 8) and France (N = 7), followed by Germany, Iran, and Spain (N=6 papers each). The remaining countries published five or fewer articles. Likewise, the GCS range from 0 to 439 (Mean = 61.33; SD = 99.91). The highest values were recorded for the United States (GCS = 439), Canada (GCS = 394), United Kingdom (GCS = 282), Australia (GCS = 195), Finland (GCS = 131) and Switzerland (GCS = 116).

The results also indicate that the authors of the selected papers were affiliated with a total of 275 institutions. The number of articles published by the various institutions ranged from 1 to 6 (Mean = 1.19; SD = 0.59). The most prolific institutions were Toronto University (Canada) (N = 6), McMaster University (Canada) (N = 5), and Monash University (Australia) (N = 4), followed by the National Institute for Health and Welfare (Finland), Shouguang People’s Hospital (China), University Jyvaskyla (Finland), and York University (Canada) (N=3 each). The remaining institutions published two or fewer related articles. For the most frequently cited institutions, citations range from 0-150 (Mean = 16.34, SD = 20.13). The four most frequently cited institutions were McMaster University (Canada) (GCS = 150), the University of Surrey (England) (GCS = 95), Stockholm University (Sweden) (GCS = 92) and York University (Canada) (GCS = 81). The remaining institutions had 79 or fewer citations. The 128 selected articles were produced by 487 researchers (Mean = 1.09; SD = 0.38). The authors Denton and Zeytinoglu each published 4 articles. The authors Burke, Elovainio, Heponiemi, Sinervo, Dong, Xu, and Zhang each published 3 articles. The other authors had between 1 and 2 publications. The number of citations of the authors ranged from 0 to 136 (Mean = 15.55; SD = 19.22). 

The 10 most cited authors were Denton (GCS = 136; GCS/t = 9.10), Zeytinoglu (GCS = 136; GCS/t = 9.10), Davies, (GCS = 114; GCS/t = 6.55), Lian (GCS = 96; GCS/t = 5.05), Hellgren, Naswall and Sverke (GCS = 92; GCS/t = 5.75 each), Burke (GCS = 81; GCS/t = 6.18), Adams and Tovey (GCS = 73; GCS/t = 3.32 each) (Table 1).

The 10 most frequently cited papers (GCS ≥ 44) are shown in Table 2.

#### 3.1.1. Co-Author Network

The network of co-authors identified 28 authors, organized in 9 groups. The groups of four authors (N = 4), three authors (N = 2) and two authors (N = 3) are shown in Figure 2, in which the thickness of the line indicates the level of collaboration. The authors who collaborate most are Denton and Zeytinoglu, followed by Elovainio, Heponiemi, and Sinervo, Zhang, Xu, and Dong. Finally, Urquhart, Kelsall, and Hoe. The institutions to which they belong in the same order of appearance are McMaster University (Denton, McMaster Centre for Gerontological Studies and Zeytinoglu, DeGroote School of Business); National Institute for Health and Welfare, Helsinki, Finland (Elovainio and Heponiemi) and Institute of Work, Health and Organisations, University of Nottingham, United Kingdom (Sinervo); Medical Department, Shouguang People’s Hospital Shouguang, China (Zhang, Xu, and Dong); Department of Epidemiology and Preventive Medicine, School of Public Health and Preventive Medicine, Monash University, Australia (Urquhart, Kelsall, and Hoe).

#### 3.1.2. Co-Citations Network

A collaborative citation is used to designate when the work of two or more authors is cited simultaneously in the same article (i.e., they are co-cited). A threshold of 2 or more collaborative citations was established in the co-citation network (Figure 3), 48 authors were identified using this criterion.

#### 3.1.3. Thematic Analysis

The keyword analysis provided 3,457 words. After reviewing duplicates and synonyms, eliminating topics that referred to the design, research methodology or other non-relevant aspects and establishing a frequency of appearance of ≥ 4 as criteria for the inclusion of words, 96 topics were obtained. Sixty percent of the most relevant topics [37] resulted in 58 concepts, which were analyzed further. These topics were grouped into five categories (Figure 4): “Working conditions related to job insecurity”, identified in red; “Job insecurity as a risk factor”, identified in blue; “Aspects related to organizational performance”, identified in purple; “Aspects related to emotional consequences of job insecurity”, identified in yellow; “Aspects related to intervention, training and prevention of job insecurity”, identified in green. As suggested by De Witte, Vander Elst and De Cuyper [6] these categories can be grouped as (a) antecedents of job insecurity (red and blue), (b) consequences of job insecurity (purple and yellow), and finally (c) interventions to reduce job insecurity (green).

### 3.2. Main Results

Most of the studies selected were quantitative studies (88%), and to a lesser extent qualitative studies (8%) and literature review studies (4%). Two types of questionnaires measuring job insecurity were used in most studies: specific measures of job insecurity, and broader measures of psychosocial risks in the work environment in general, of which insecurity was only one factor or dimension (multi-dimensional) [5]. Specific measures include the Job Insecurity Scales of Ashford, Lee and Bobko [38], Hellgren, Sverke and Isaksson [39], and Vander Elst et al. [40] (N = 3 papers, each).

The most widely used multi-dimensional measures were in descending order: the Job Content Questionnaire (JCQ) [41] (N = 12 papers); the Effort-Reward Imbalance Questionnaire (ERI) [42] (N = 4 papers); the Copenhagen Psychosocial Questionnaire (COPSOQ) [43] (N = 3 papers) and its short version (N = 1 article).

#### 3.2.1. Causes and Consequences of Job Insecurity

In the studies analyzed, 10% of the papers deal with the causes of job insecurity: temporary employment, job instability, changes in working environments, working conditions (workload, working hours, organization of the work process, labor rights). The remainder (90%) focus on the consequences of job insecurity. Three categories of consequences of job insecurity can be observed: (a) health (musculoskeletal disorders, increased risk of coronary pathologies), well-being (burnout, work engagement, including motivation and empowerment) and work-related attitudes (job satisfaction, commitment) (b) behaviors (influence on the decision to become a nurse, leaving the profession, turnover and emigration), and (c) variables moderating the consequences of insecurity (personality, resilience, work dedication and age, emotional intelligence, positive leadership, and adequate management).

#### 3.2.2. In-Depth Analysis of Papers in Which Job Insecurity is a Central Topic

Next, for a more in-depth study of job insecurity, we selected the papers in which job insecurity is a central topic (13, marked with * in the reference list). Most studies focused on the consequences of job insecurity in nursing (N = 7 papers): increased migration to other countries to find better work conditions [19], increased psychosocial stress [30], musculoskeletal disorders and stress [44]. Others refer to negative work attitudes, and reduced satisfaction, psychological well-being, and hospital performance [45]. Job insecurity was positively associated with depression and anxiety [46], with cardiovascular indicators such as blood pressure and cardiovascular symptoms [47], and with the increased short-term risk of suffering a heart attack in women [48].

Some papers analyzed the background to job insecurity, suggesting that aspects such as poor working conditions (limited continuing education activities, high-risk exposure, low work satisfaction, low participation in the organization of activities, poor ergonomics, poor information about outcomes, limited social recognition and support from immediate superiors), high levels of work intensity and inadequate work management processes could increase perceptions of job insecurity, and may contribute to errors in care provision [49].

A final set of papers analyzed the moderating effect of specific variables on job insecurity and its causes or outcomes. Emotional intelligence moderated the relationship between job insecurity and somatic complaints [50]. The ability to deal with emotions and relationships with supervisors was an important resource which buffered the negative consequences of job insecurity [50]. Personality characteristics (negative affectivity, positive affectivity, and external locus of control) also moderated the relationship between job insecurity and outcomes (mental health complaints, job dissatisfaction, and work-induced stress) [51]. Negative affectivity exacerbated mental health complaints and job-induced stress as a result of job insecurity, whereas positive affectivity buffered these outcomes. An external locus of control strengthened the associations of job insecurity with mental health complaints, job dissatisfaction, and job-induced tension. In addition, positive leadership and fair management alleviated the impact of job insecurity when working in a changing and uncertain environment [52]. Meanwhile, other studies reported that being younger and with less dedication to the job provided protection against the negative effects of high job insecurity on parental satisfaction [8]. Finally, the study by Sarwar, Naseer and Zhong [53] showed that job insecurity mediated the association of workplace bullying with deviant work behaviors in nurses. This indirect effect was stronger when resilience and supervisor support was low.

## 4. Discussion

The analysis of scientific production is an essential tool for evaluating knowledge, raising research questions and determining the progress of a discipline. These aspects applied to the context of job insecurity in the nursing lead to some interesting findings. According to the bibliometric indicators, the most significant growth occurred in 2013, with 11 published papers, which came from 157 different countries. The most productive countries were the United States, Canada, Australia, Finland, and the United Kingdom. The most prolific institutions were Toronto University (Canada), McMaster University (Canada) and Monash University (Australia). The 128 selected papers were produced by 487 researchers. The most prolific authors were Denton, Zeytinoglu, Burke, Elovainio, Heponiemi, Sinervo, Dong, Xu, and Zhang. The most cited article was written by Denton and Zeytinoglu (2002), which received 96 citations, and was written by the most prolific and extensively cited authors. The second most cited article was written by Näswall, Sverke and Hellgren (2005) with 92 citations, who are among the most cited authors, and the third article was by Tovey and Adams (1999) with 73 citations, published in the Journal of Advanced Nursing, who are among the most cited authors and appeared in the most prolific journal. Only two of the most frequently cited articles focus on job insecurity as such (articles 2 and 4). However, article 2 analyses the impact of personality in a sample of nurses, while article 4 does not deal specifically with nurses or health workers when discussing the effects of job insecurity. The remaining articles are more general assessments of the well-being and job characteristics of nurses, without a specific focus on job insecurity. When discussing insecurity, the authors consider it as ‘just one’ concept within a general analysis of labor risks or working conditions. This suggests that there are virtually no studies that specifically analyze the effects of job insecurity among nurses as a specific target group. The most collaborative authors are distributed in four groups: Denton and Zeytinoglu, followed by Elovainio, Heponiemi and Sinervo, then Zhang, Xu and Dong, and finally, Urquhart, Kelsall, and Hoe. The most collaborative authors are also the most prolific and with the most cited articles. Nevertheless, there are no authors with many publications, and even the most productive and the most cited authors have not produced many articles. All these results might suggest that job insecurity in nursing is an emerging topic, with a little academic background within the nursing field. The most widely used multi-dimensional measure was the Job Content Questionnaire (JCQ) [41]. The JCQ is a validated instrument used to measure the social and psychological structure of work. This instrument has 49 items, including latitude of decision (the worker’s control over their own work), discretion ability subscales, psychological requirements, supervisor and social support coworker subscales, physical demands of the job, and job insecurity. The Likert-type response scales ranged from 1 (strongly disagree) to 4 (strongly agree). It has been widely used to examine psychosocial stress factors in various work environments with appropriate psychometric properties [54].

The main results of this study are consistent with the results of the general literature on job insecurity, which reports negative correlations with job satisfaction, mental well-being, and physical health [5,7]. Job insecurity also correlates with sleep disorders [55], increased blood pressure [10], increased likelihood of heart disease [56], and with the increased short-term risk of suffering from a myocardial infarction [35]. Job insecurity is also related to the employment contract, with temporary employees experiencing more job insecurity than employees with a permanent contract [57]. Job insecurity is related to poorer scores for the various dimensions of burnout [58] and reduced engagement at work [59,60]. It is mostly measured with more broad-based instruments to assess psychosocial risks in a multi-dimensional way. This suggests that job insecurity is considered to be ‘just one of many’ occupational hazards, rather than a central issue.

These findings represent some progress in the international nursing field, as bibliometric studies on job insecurity have been lacking to date. Job insecurity is also a crucial aspect for consideration given its consequences, especially among nurses, including increased migration to other countries to find better working conditions [19], psychosocial stress [30], depression and anxiety [46], musculoskeletal disorders [44], blood pressure and cardiovascular symptoms [47], and poorer performance in the hospital [45]. 

This study is not without limitations, as indicators from a single database were used. Other types of indexing or journals are needed in the future. A more extensive analysis, incorporating other databases would therefore be interesting in future research. Most studies included have limitations that hinder the generalizability of the results (a specific context or region, convenience sampling, mixed sample). Perhaps it would be worthwhile placing the analyzed problems in a cultural context and without mixed samples in the future. Despite these limitations, the results are of particular interest since they summarize the available evidence in the area and present some gaps in the literature. This will help establish healthy working environments, favoring the retention of nurses and the improvement of the service provided to patients [61,62]. 

According to Schaufeli [63], future nursing research on job insecurity should focus more on uncovering the psychological mechanisms underlying the subjective experience of insecurity, and could study the consequences of job insecurity in different social security systems and based on re-employment rates. Healthcare managers can use the results to design training programs and interventions to improve working conditions. They can focus on increasing control at work, promoting open and timely communication about future organizational plans [64], increasing the participation of nurses in decision making about the future of the organization [65], and improving the employability of workers [66].

## 5. Conclusions

This study shows that job insecurity is a real problem for nurses and the world’s health systems, with a significant impact on their health and well-being. The main bibliometric indicators show a growth in this topic in nursing since 2013. The most productive and widely cited authors are Denton and Zeytinoglu. The most productive universities are Toronto University, McMaster University, and Monash University. The most productive countries are the United States, Canada, Australia, Finland, and the United Kingdom. The most widely used multi-dimensional measure was Karasek’s Job Content Questionnaire (JCQ). Finally, job insecurity has consequences for health, well-being, and work-related attitudes, behaviors. However, as an emerging topic, job insecurity is rarely studied in specialized literature. We consequently recommend extending this research tradition to the nursing context.

These results can be considered an initial approach to the study of job insecurity in nursing, covering the causes and consequences of job insecurity as well as the moderating variables of job insecurity (emotional intelligence, personality characteristics, positive leadership, and fair management). These moderating variables suggest several ways which organizations and policy makers can intervene to address job insecurity among their nurses.

## Figures and Tables

**Figure 1 ijerph-18-00663-f001:**
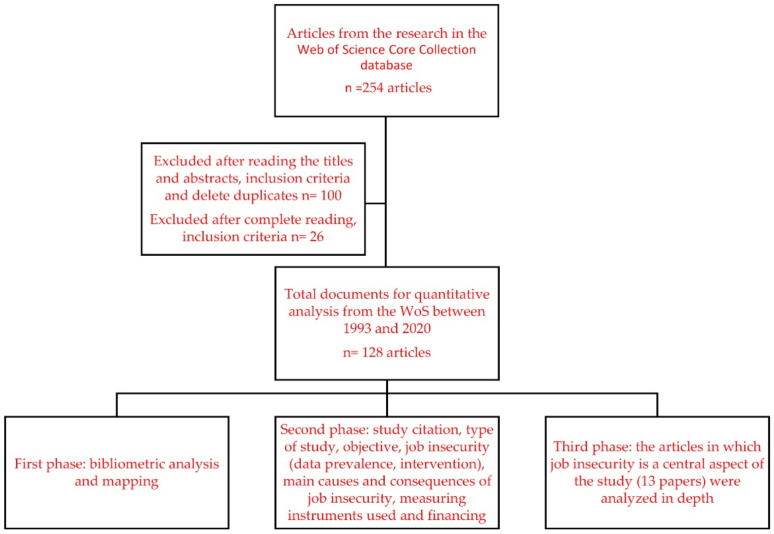
Selection of articles and research design description.

**Figure 2 ijerph-18-00663-f002:**
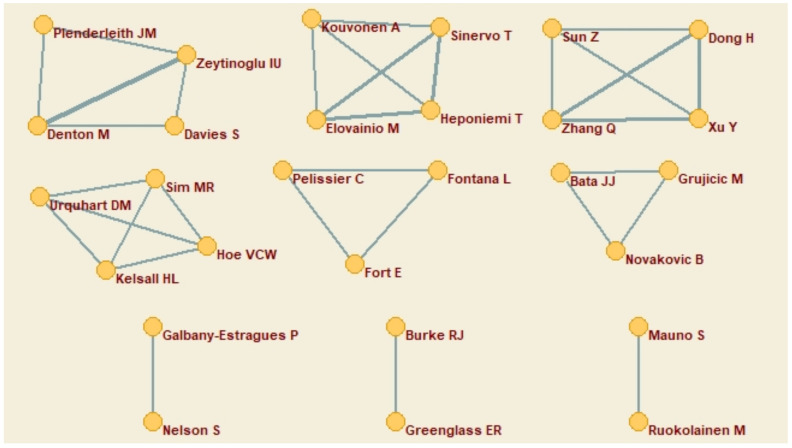
Co-author network.

**Figure 3 ijerph-18-00663-f003:**
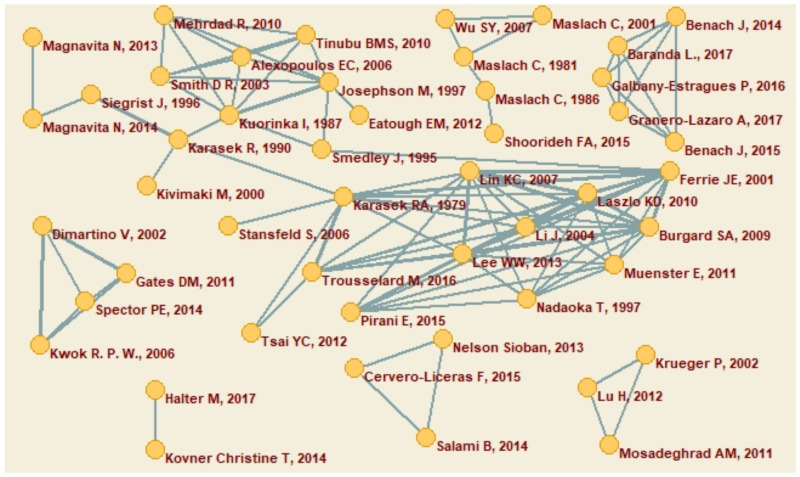
Co-citations network (≥2 co-citations of publications).

**Figure 4 ijerph-18-00663-f004:**
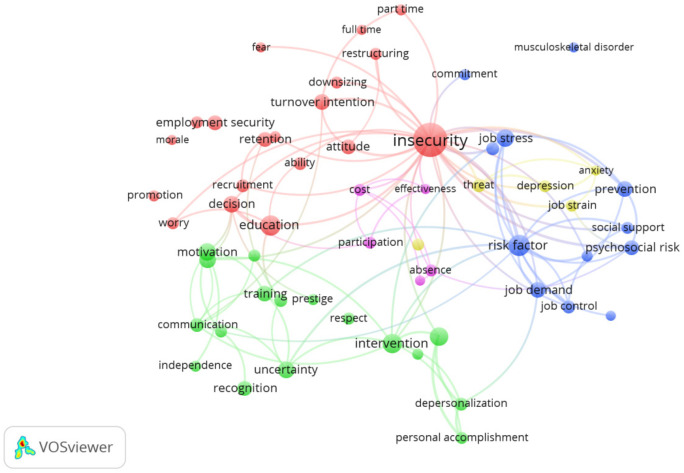
Topics network.

**Table 1 ijerph-18-00663-t001:** Most cited authors (GCS ≥ 54).

AUTHOR	PAPERS	%	GCS	GCS/T
Denton M.	4	3.1	136	9.02
Zeytinoglu I. U.	4	3.1	136	9.02
Davies S.	2	1.6	114	6.55
Lian J.	1	0.8	96	5.05
Hellgren J.	1	0.8	92	5.75
Naswall K.	1	0.8	92	5.75
Sverke M.	1	0.8	92	5.75
Burke R.J.	3	2.4	81	6.18
Adams A. E.	1	0.8	73	3.32
Tovey E. J.	1	0.8	73	3.32
Berkman L. F.	1	0.8	70	4.12
Colditz G. A.	1	0.8	70	4.12
Kawachi I.	1	0.8	70	4.12
Lee S.	1	0.8	70	4.12
Greenglass E. R.	2	1.6	62	3.01
Barling J.	1	0.8	54	3.18
McNeese-Smith D. K.	1	0.8	54	2.7
Taylor B.	1	0.8	54	3.18

Note: GCS = Global Citation Score in the Clarivate Analytics´ database Web of Science; GCS/T = GCS per year.

**Table 2 ijerph-18-00663-t002:** The 10 most cited papers (GCS ≥ 44).

PAPERS	GCS	GCS/T
Denton, M.; Zeytinoglu, I.U.; Davies, S.; Lian, J. Job stress and job dissatisfaction of home care workers in the context of health care restructuring. *Int. J. Health Serv.* **2002**, *32*, 327–357.	96	5.05
2. Näswall, K.; Sverke, M.; Hellgren, J. The moderating role of personality characteristics on the relationship between job insecurity and strain. *Work Stress* **2005**, *19*, 37–49.	92	5.75
3. Tovey, E.J.; Adams, A.E. The changing nature of nurses’ job satisfaction: an exploration of sources of satisfaction in the 1990s. *J. Adv. Nurs.* **1999**, *30*, 150–158.	73	3.32
4. Lee, S.; Colditz, G.A.; Berkman, L.F.; Kawachi, I. Prospective study of job insecurity and coronary heart disease in US women. *Ann. Epidemiol.* **2004**, *14*, 24–30.	70	4.12
5. McNeese-Smith, D.K.; Nazarey, M. A nursing shortage: Building organizational commitment among nurses/practitioner application. *J. Healthc. Manag.* **2001**, *46*, 173–186.	54	2.70
6. Taylor, B.; Barling, J. Identifying sources and effects of carer fatigue and burnout for mental health nurses: A qualitative approach. *Int. J. Mental Health Nurs.* **2004**, *13*, 117–125.	54	3.18
7. Eriksen, W.; Tambs, K.; Knardahl, S. Work factors and psychological distress in nurses’ aides: a prospective cohort study. *BMC Public Health* **2006**, *6*, 290.	47	3.13
8. Lyndon, A.; Sexton, J.B.; Simpson, K.R.; Rosenstein, A.; Lee, K.A.; Wachter, R.M. Predictors of likelihood of speaking up about safety concerns in labour and delivery. *BMJ Qual. Saf.* **2012**, *21* 791–799.	47	5.22
9. Laschinger, H.K.S.; Purdy, N.; Cho, J.; Almost, J. Antecedents and consequences of nurse managers’ perceptions of organizational support. *Nurs. Econ.* **2006**, *24*, 20–29.	44	2.93
10. Yoon, S.L.; Kim, J.H. Job-related stress, emotional labor, and depressive symptoms among Korean nurses. *J. Nurs. Scholarsh.* **2013**, *45*, 169–176.	43	5.38

Note: GCS = Global Citation Score of the Clarivate Analytics´ database Web of Science; GCS/t = GCS per year.

## Data Availability

The data presented in this study are available on request from the corresponding author. The data are not publicly available due to its huge size.

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
