# Peer review of "Job Insecurity in Nursing: A Bibliometric Analysis"

_ijerph, 2021, doi:10.3390/ijerph18020663_

Round 1
Reviewer 1 Report
It is always refreshing to read an excellent paper. The introduction and the objectives are very clear and explicit. Having taken part in a scoping study (similar methodology) myself in the last 18 months, I can say that their methodology is state-of-the-art. However I must agree with the authors that " a more extensive analysis, incorporating other databases, would therefore be interesting in future research".
The analyses are well performed, and the network maps are quite an asset. Their results are interesting and well presented.
My only regret is about the conclusion: only two short paragraphs. One would expect the authors to expand a little more on the implications of this research and on their recommendations on the basis of this research and its results. The rich findings they have come up with could take them a little further.
Needless to say, their bibliography is up to date and solid.
Author Response
My only regret is about the conclusion: only two short paragraphs. One would expect the authors to expand a little more on the implications of this research and on their recommendations on the basis of this research and its results. The rich findings they have come up with could take them a little further.
Needless to say, their bibliography is up to date and solid.
Dear reviewer,
We would like to thank you for their time and effort in revising our paper. We really appreciate the suggestions made and we believe the document has greatly benefited from it. The paper has been reedited by a native English speaker. All changes have been included in the article in red
Response reviewer 1 comments:
We have modified the conclusions according the reviewer’s suggestions, with the aim of responding to the objectives of the study as well as eliminating references and improving the implications of the study. Page 11 L 373-388.
- Conclusions
This study shows that job insecurity is a real problem for nurses and the world's health systems, with a significant impact on their health and well-being. The main bibliometric indicators show a growth in this topic in nursing since 2013. The most productive and widely cited authors are Denton and Zeytinoglu. The most productive universities are Toronto University, McMaster University and Monash University. The most productive countries are the United States, Canada, Australia, Finland, and the United Kingdom. The most widely used multi-dimensional measure was Karasek’s Job Content Questionnaire (JCQ). Finally, job insecurity has consequences for health, well-being and work-related attitudes, behaviours. However, as an emerging topic, job insecurity is rarely studied in specialized literature. We consequently recommend extending this research tradition to the nursing context.
These results can be considered an initial approach to the study of job insecurity in nursing, covering the causes and consequences of job insecurity as well as the moderating variables of job insecurity (emotional intelligence, personality characteristics, positive leadership and fair management). These moderating variables suggest several ways which organizations and policy makers can intervene to address job insecurity among their nurses
Reviewer 2 Report
It is an indisputable fact that insecurity at work is an occupational stressor, with negative consequences for an individual's physical, mental and social health. Hence, the authors took up a very important and topical problem. The publication emphasizes that nurses constitute the largest group of health professionals (59%) and play a key role in global health systems. Among the many psychosocial risks, job insecurity plays a significant role. This study analyses the state of research on job insecurity in nursing using a bibliometric analysis.
Bibliometric methods are used, inter alia, when the overwhelming amount of literature in a given field does not allow for its synthetic analysis, for capturing the main trends of development in a given area and for determining the relationship between them. Assuming that a network of connections between scientific publications it is inherently cognitive, relations are analysed between citations of publications in a given field.
This publication assumes that the existing literature on job insecurity among nurses is scarce and that a bibliometric study on nursing appears to be insufficient. It is a pity that the research covers such a short period of time - it would be extremely interesting to learn about the tendencies of interest in the problem of job insecurity among nurses over the period of, for example, 60-70 years. Have (how?) these tendencies changed?
When analysing the most cited publications, it is worth establishing a network of connections between them. This was done (3.1.1) - in terms of collaboration between researchers and in terms of citations (3.1.2). The presentation of thematic analysis (Fig. 3- topics network) is very interesting. Another very good visualization tool for links is the historiographer, showing the cross-reference network between publications in a local database. By analysing the historiographer, one could find out the roots of research into job insecurity among nurses. Perhaps in the future the authors will decide to implement this approach.
These are minor comments and they do not lower the evaluation of the study. The work is not very revealing, but it is very useful. Perhaps in the future it is worth placing the analysed problems in a cultural context. Different countries struggle with problems in the health care systems, and solving them most often depends on the adopted employment and development policy (hence the differences: impact, the most prolific institutions and countries, citation networks).
The research plan is well constructed but could still be improved.
I hope that the authors will continue their research in the context of the epidemiological threat of COVID-19. This context has significantly influenced the subjective sense of security among nurses. It is extremely interesting whether the authors nowadays still consider the topic of job insecurity as requiring extensive research, or whether other problems turn out to be more important (e.g. lack of recognition, job risk, working conditions).
Author Response
Dear reviewer,
We would like to thank you for their time and effort in revising our paper. We really appreciate the suggestions made and we believe the document has greatly benefited from it. The paper has been reedited by a native English speaker. All changes have been included in the article in red
Response reviewer 2 comments:
This publication assumes that the existing literature on job insecurity among nurses is scarce and that a bibliometric study on nursing appears to be insufficient. It is a pity that the research covers such a short period of time - it would be extremely interesting to learn about the tendencies of interest in the problem of job insecurity among nurses over the period of, for example, 60-70 years. Have (how?) these tendencies changed?
Response
The research is not limited in time. Web of Science Core Collection database using the SCI-EXPANDED and SSCI indexes on 6 March 2020. One of the inclusion criteria was articles published in any language in all years (Page 3 L 111).
When analysing the most cited publications, it is worth establishing a network of connections between them. This was done (3.1.1) - in terms of collaboration between researchers and in terms of citations (3.1.2). The presentation of thematic analysis (Fig. 3- topics network) is very interesting. Another very good visualization tool for links is the historiographer, showing the cross-reference network between publications in a local database. By analysing the historiographer, one could find out the roots of research into job insecurity among nurses. Perhaps in the future the authors will decide to implement this approach.
Response:
Thank you very much for your suggestions and valuable comments, we will take them into account in future research.
These are minor comments and they do not lower the evaluation of the study. The work is not very revealing, but it is very useful. Perhaps in the future it is worth placing the analysed problems in a cultural context. Different countries struggle with problems in the health care systems, and solving them most often depends on the adopted employment and development policy (hence the differences: impact, the most prolific institutions and countries, citation networks).
Response:
We really appreciate the suggestions made on this regard, we have included at future research the analysed problems in a cultural context. Page 10 L 351-354
“Most studies included have limitations that hinder the generalizability of the results (a specific context or region, convenience sampling, mixed sample). Perhaps it would be worthwhile placing the analysed problems in a cultural context and without mixed samples in the futureThe research plan is well constructed but could still be improved”.
I hope that the authors will continue their research in the context of the epidemiological threat of COVID-19. This context has significantly influenced the subjective sense of security among nurses. It is extremely interesting whether the authors nowadays still consider the topic of job insecurity as requiring extensive research, or whether other problems turn out to be more important (e.g. lack of recognition, job risk, working conditions).
Response:
Thank you very much for your suggestions and valuable comments, will continue our research in the context of the epidemiological threat of COVID-19. The authors still consider the issue of job insecurity as something that requires extensive research.
Reviewer 3 Report
Thank you to the authors for the manuscript. Unfortunately there are three critical issues in the manuscript.
The authors have not clearly and convincingly justified the need for this study. This is a critical issue in the manuscript.
There is no literature review. A literature review is a summary and analysis of the research to date for a construct, theory, etc.
Also, what are the practical implications based on findings of this study? More specific and realistic (substantial) implications are required. It is difficult to recognize difference from the already-preceded research. In addition, even the supplementation is necessary for research limitations and future research.
Because of these critical issues, I can’t recommend publication. I strongly encourage the authors to rethink the abstract, justification, literature, and theoretical underpinning of the study. Also, this examinant proposed that the result of this study be inserted into research note rather full-paper.
Author Response
Dear reviewer,
We would like to thank you for their time and effort in revising our paper. We really appreciate the suggestions made and we believe the document has greatly benefited from it.
In general, introduction, research design, methods, discussion and conclusions have been modified with more updated references and completed the background of the study. References have been reviewed, some of them have been adapted in order to comply with their comments and the journal´s requirements. The paper has been reedited by a native English speaker. All changes have been included in the article in red.
Response reviewer 3 comments:
Thank you to the authors for the manuscript. Unfortunately there are three critical issues in the manuscript.
The authors have not clearly and convincingly justified the need for this study. This is a critical issue in the manuscript.
There is no literature review. A literature review is a summary and analysis of the research to date for a construct, theory, etc.
Response:
We really appreciate the suggestions made on this regard, they have been very helpful. The introduction section has been modified to improve the conceptualization of job insecurity and its theoretical foundation. The importance of job insecurity has been exposed. The background in general and specifically in nursing has been explained. Moreover, relevant references have been included. Page 1-2, L 29-82.
- Introduction
"The restructuring of labour markets, economic crisis and globalization, and increasingly excessive labour demands on workers, (flexible contracts, reduced staffing, temporary contracts, increased workload and pressure, and poor work-life balance), can become labour stressors throughout the world in both industrialized and developing countries [1]. However, work-related stress is determined by multiple causes [2], including psychosocial risk factors, namely work design characteristics related to the general context of the organization that may impair the psychological and/or physical well-being of employees [3]. Among the psychosocial risks that seem to negatively affect workers, job insecurity is one which has the most impact on workers and on society in general [4]. Job insecurity can be defined as a concern about continuing future employment [5] and is a subjective perception: workers may perceive the same situation in slightly different ways due to their personality or position in the labour market [6]. Job insecurity is a work stressor, with negative consequences for the individual's physical, psychological and social health [6-7], and their family [8]. Theories such as the Job Demand Control Model [9] contribute to the interpretation of job insecurity. The Job Demand Control Model predicts that low levels of job control, high demands at work and the interaction between them lead to a higher risk of strain, including poorer health and well-being [10]. Job insecurity is described in terms of lack of control, and implies a lack of resources, especially control [10]. According to the Organisation for Economic Co-operation and Development (OECD), job insecurity is the most important aspect of work in almost all countries, and the European Social Survey (ESS) and the International Social Survey Programme (ISSP) confirm this perception among workers [11]. In addition, job insecurity not only has adverse effects on people's health and well-being; it also negatively affects employees’ job satisfaction and commitment, and reduces their work-related health and well-being [4,12]. Job insecurity also reduces the satisfaction of basic human needs [9], performance at work, and creativity [13-14]. Furthermore, it can affect companies’ performance by reducing worker retention rates, investment in company-specific skills and productivity, and also impacts on society at large, social unrest, consumer confidence and savings behaviour. For these reasons, insecurity is one of the three main dimensions in the new OECD framework for the measurement and assessment of the quality of employment [15].
These aspects are crucial to any type of profession but are paramount for nurses, because nurses are the largest group of health professionals (59%) and play a vital role in global health systems [16]. The contribution of nurses to global health is undisputed, and investing in improving their quality of life benefits society [17]. Job insecurity among nurses has also been associated with migration to other countries [18] in search of higher salaries and a better quality of life. It has also been linked to worsening health conditions for nurses and increasing burnout, stress and vulnerability [19-20]. The increase in nurses’ workload increases the probability of hospital deaths, and significantly reduces the quality of care [21] and holistic nursing care [22]. Job insecurity makes daily work and therapeutic relationships difficult [23] because professionals are unable to provide quality care. Improved working conditions and professional development affects not only their well-being and quality of life, but also their performance and by extension the functioning of the entire healthcare system [24]. As a result, adequate staffing and prioritization of occupational health and safety is essential, as the WHO suggests [16].
Despite its importance, the decline in the average employment tenure of people aged 30-50 in all countries between 1992 and 2014 is evidence of increasing job insecurity [25]. The existing literature on this subject is scarce, and a bibliometric study on the subject in nursing appears to be lacking. For this reason, this study analyzes the state of research on job insecurity in nursing using a bibliometric analysis in order to determine the most important findings in the literature in this field, taking into account the distribution of publications, authorship, co-authorship, impact, the most prolific institutions and countries, citation networks, and the most relevant topics about job insecurity in nursing [26]. It also aims to establish the most widely used instruments for measuring job insecurity, and the main causes and consequences of job insecurity among nurses [27]. As a result of the lack of studies of this type, we believe that it could be very useful for health policymakers and health managers in all countries and regions to design programs to retain nurses and ensure the sustainability of health care systems [18], improve nursing processes and working conditions [28]."
Also, what are the practical implications based on findings of this study? More specific and realistic (substantial) implications are required. It is difficult to recognize difference from the already-preceded research. In addition, even the supplementation is necessary for research limitations and future research.
Response:
We really appreciate the suggestions made on this regard. The practical implications, limitations and future research have been modified according to the reviewer suggestions. Page 10 and 11 L 358-388
"According to Schaufeli [62], future nursing research on job insecurity should focus more on uncovering the psychological mechanisms underlying the subjective experience of insecurity, and could study the consequences of job insecurity in different social security systems and based on re-employment rates. Healthcare managers can use the results to design training programs and interventions to improve working conditions. They can focus on increasing control at work, promoting open and timely communication about future organizational plans [63], increasing the participation of nurses in decision making about the future of the organization [64], and improving the employability of workers [65].
- Conclusions
This study shows that job insecurity is a real problem for nurses and the world's health systems, with a significant impact on their health and well-being. The main bibliometric indicators show a growth in this topic in nursing since 2013. The most productive and widely cited authors are Denton and Zeytinoglu. The most productive universities are Toronto University, McMaster University and Monash University. The most productive countries are the United States, Canada, Australia, Finland, and the United Kingdom. The most widely used multi-dimensional measure was Karasek’s Job Content Questionnaire (JCQ). Finally, job insecurity has consequences for health, well-being and work-related attitudes, behaviours. However, as an emerging topic, job insecurity is rarely studied in specialized literature. We consequently recommend extending this research tradition to the nursing context.
These results can be considered an initial approach to the study of job insecurity in nursing, covering the causes and consequences of job insecurity as well as the moderating variables of job insecurity (emotional intelligence, personality characteristics, positive leadership and fair management). These moderating variables suggest several ways which organizations and policy makers can intervene to address job insecurity among their nurses."
Because of these critical issues, I can’t recommend publication. I strongly encourage the authors to rethink the abstract, justification, literature, and theoretical underpinning of the study. Also, this examinant proposed that the result of this study be inserted into research note rather full-paper.
Response
In general, introduction, research design, methods, discussion and conclusions have been modified with more updated references and completed the background of the study according to the reviewers' comments. We appreciate your comments and consider that they have contributed to improve the article, taking into account the relevance of the results obtained to be inserted in a full-paper for the reasons explained in the study (Page 2 L 74-82):
“For this reason, this study analyzes the state of research on job insecurity in nursing using a bibliometric analysis in order to determine the most important findings in the literature in this field, taking into account the distribution of publications, authorship, co-authorship, impact, the most prolific institutions and countries, citation networks, and the most relevant topics about job insecurity in nursing [26]. It also aims to establish the most widely used instruments for measuring job insecurity, and the main causes and consequences of job insecurity among nurses [27]. As a result of the lack of studies of this type, we believe that it could be very useful for health policymakers and health managers in all countries and regions to design programs to retain nurses and ensure the sustainability of health care systems [18], improve nursing processes and working conditions [28]”.
Reviewer 4 Report
Dear authors
Although the study is well-written and easy to read I think that the study results (most productive authors, universities, journals, most cited studies or keywords and their relations etc.) is not a key topic for nurses.
Also, the selection process it has some mistakes. For example, one study included in the list as "most cited studies" do not have nurses in their sample (Number 10 it has nurse auxiliaries) and another one (Number 1) has a mixed sample withouth independent results for nurses.
I do not understand why you have not used databases like Pubmed, CINAHL or Scopus in an bibliometric analysis about nursing topics.
Some parts of the conclusion are not based on the results. The conclusion should not use references to other manuscripts, it must contain only the response to the study aims based on your results.
Kind regards
Author Response
Dear reviewer,
We would like to thank you for their time and effort in revising our paper. We really appreciate the suggestions made and we believe the document has greatly benefited from it.
In general, introduction, research design, methods, discussion and conclusions have been modified with more updated references and completed the background of the study. References have been reviewed, some of them have been adapted in order to comply with their comments and the journal´s requirements. The paper has been reedited by a native English speaker. All changes have been included in the article in red.
Response reviewer 4 comments:
Although the study is well-written and easy to read I think that the study results (most productive authors, universities, journals, most cited studies or keywords and their relations etc.) is not a key topic for nurses.
Response:
We appreciate your concern, the authors agree with the reviewer's comment. The bibliometric indicators are not a key topic for nurses but help us explore publishing patterns and trends, describe concept development, new emerging areas of research, research gaps, and information about and characteristics of the existing literature and recent advances in key topics.
Also, the selection process it has some mistakes. For example, one study included in the list as "most cited studies" do not have nurses in their sample (Number 10 it has nurse auxiliaries) and another one (Number 1) has a mixed sample withouth independent results for nurses.
Response:
We really appreciate the suggestions made on this regard, have been very helpful. The article 1 has a mixed sample involving nurses, there are nurses´ focus group results (attached below). We have included these comments as a limitations and future research (Page 10 L 351-354).
“Most studies included have limitations that hinder the generalizability of the results (a specific context or region, convenience sampling, mixed sample). Perhaps it would be worthwhile placing the analysed problems in a cultural context and without mixed samples in the future”
Article 1
“Focus group participants indicated that health care restructuring has resulted in organizational change, budget cuts, heavier workloads, job insecurity, loss of organizational support, loss of peer support, and loss of time to provide emotional laboring, or the “caring” aspects of home care work. Analyses of survey data show that organizational change, fear of job, loss, heavy workloads, and lack of organizational and peer support lead to increased job stress and decreased levels of job satisfaction.
Focus Group Results
Many home care workers feared that they might lose their jobs due to the restructuring of home health care delivery. A coordinator expressed these fears as follows: “I worked here for quite a while and I thought my job was stable and, like you said, we work for a good organization, it’s been around for almost 100 years, and so now you can’t even think, well next year will I have a job?” Nurses also felt job insecurity: “I think there is a little more insecurity and fear of job security, nobody really knows, everybody thinks their own job classification may be in jeopardy and nobody is really sure.”
In study 10, the sample includes nursing assistants and care assistants, we agree on the selection error and apologize, it has been corrected in the table.
I do not understand why you have not used databases like Pubmed, CINAHL or Scopus in an bibliometric analysis about nursing topics.
Response:
We really appreciate the suggestions made on this regard. The authors have introduced the use of one database as a study limitation. Page 10 L 349-351.
“This study is not without limitations, as indicators from a single database were used. Other types of indexing or journals are needed in the future. A more extensive analysis, incorporating other databases would therefore be interesting in future research”.
We used the Web of Science because is considered a complete database. The journal impact factor (JIF) published by Thompson Reuters via Journal Citation Reports (JCR®) has been the ‘gold standard’ of ranking journals and the impact this has on where nursing academics and researchers publish (Polit & Northam 2011, Hunt et al. 2012).
Polit D.F. & Northam S. (2011) Impact factors in nursing journals. Nursing Outlook 59, 18–28.
Hunt G.E., Happell B., Chan S.W. & Cleary M. (2012) Citation analysis of mental health nursing journals: How should we rank thee? International Journal of Mental Health Nursing 21, 576–580
Some parts of the conclusion are not based on the results. The conclusion should not use references to other manuscripts, it must contain only the response to the study aims based on your results.
Response:
We appreciate your concern, the authors agree with the reviewer's comment. We have modified the conclusion section without references and the response to the study aims. Page 11 L 373-388
- Conclusions
This study shows that job insecurity is a real problem for nurses and the world's health systems, with a significant impact on their health and well-being. The main bibliometric indicators show a growth in this topic in nursing since 2013. The most productive and widely cited authors are Denton and Zeytinoglu. The most productive universities are Toronto University, McMaster University and Monash University. The most productive countries are the United States, Canada, Australia, Finland, and the United Kingdom. The most widely used multi-dimensional measure was Karasek’s Job Content Questionnaire (JCQ). Finally, job insecurity has consequences for health, well-being and work-related attitudes, behaviours. However, as an emerging topic, job insecurity is rarely studied in specialized literature. We consequently recommend extending this research tradition to the nursing context.
These results can be considered an initial approach to the study of job insecurity in nursing, covering the causes and consequences of job insecurity as well as the moderating variables of job insecurity (emotional intelligence, personality characteristics, positive leadership and fair management). These moderating variables suggest several ways which organizations and policy makers can intervene to address job insecurity among their nurses.
Round 2
Reviewer 3 Report
The theoretical background needs to be supplemented more.Happy new year!
Author Response
Dear reviewer,
We would like to thank you for their time and effort in revising our paper. We really appreciate the suggestions made and we believe the document has greatly benefited from it. The paper has been reedited by a native English speaker (a certificate is attached). All changes have been included in the article in red.
Response reviewer 3 comments:
The theoretical background needs to be supplemented more.
Response
We really appreciate the suggestions made on this regard. The background has been supplemented, the Karasek's Job Demand Control Model contribute to the interpretation of job insecurity and has been explained in more detail. Page 1-2 L 43-67.
“Theories such as the Karasek's Job Demand Control Model [9] contribute to the interpretation of job insecurity. This model explains job stress in terms of the balance between the psychological job demands and the worker's level of control over them, i.e. job control [9]. Psychological job demands include role conflict, workload, role ambiguity, job insecurity and cognitive demands, among others and are the psychological stressors in the work environment. Job control involves the opportunities to develop one's own skills, and the autonomy that the work provides, i.e. the resources that the employee has. The development of these skills has a twofold aspect; on the one hand obtaining and improving sufficient capacity to carry out the tasks, and on the other hand, the possibility of working on (or carrying out) what one knows best (creative and varied work). Autonomy refers to the ability to decide on one's own tasks and control over breaks and the pace of work [9]. According to Kasarek, the Job Demand Control Model predicts that low levels of job control, high demands at work and the interaction between them lead to a higher risk of strain, including poorer health and well-being [10]. Job insecurity is described in terms of lack of control, and implies a lack of resources, especially control [10]. In this model, the health or well-being of the worker will depend on the balance between the demands of the job and the worker's own available resources. The authors Johnson and Hall introduced social support as the third dimension of this model, by establishing the job demand-control-social support model, operating in two ways [11] since workers exposed to high demands, little control and under social support present twice the risk of morbidity and mortality from cardiovascular disease than those with low-demand jobs, much control and strong social support. This dimension of social support refers to all possible levels of on-the-job interaction, both with peers and superiors.
Job insecurity is the most important aspect of work in almost all countries according to the Organisation for Economic Co-operation and Development (OECD), and the the European Social Survey (ESS) and the International Social Survey Programme (ISSP) confirm this perception among workers [12].”
A new updated citation on the consequences of job insecurity for nurses has been included. Page 2 L 81. “It has also been linked to worsening health conditions for nurses and increasing burnout, stress and vulnerability [20-21], lower job satisfaction levels [22].”
The research design must be improved.
Response:
The research design information has been reorganized to improve the whole section 2. Materials and Methods and a new figure (Figure 1), has been included to facilitate understanding of the research design. Page 3-4 L 101-156.
Thank you very much for your comments. We believe that the document has benefited from the insight of the editors and reviewers and we really appreciate your time and effort. The authors wish you a happy new year.

Reviewer 4 Report
Dear authors
Thank you for adressing my suggestions.
Kind regards
Author Response
Dear reviewer,
We would like to thank you for their time and effort in revising our paper. We really appreciate the suggestions made and we believe the document has greatly benefited from it. The paper has been reedited by a native English speaker (a certificate is attached). All changes have been included in the article in red.
Thank you very much for your comments. We believe that the document has benefited from the insight of the editors and reviewers and we really appreciate your time and effort. The authors wish you a happy new year.
